# Validation of N-Methylpyridinium as a Feasible Biomarker for Roasted Coffee Intake

Beate Brandl [1], Coline Czech [2,3], Susanne I. Wudy [4], Anja Beusch [2], Hans Hauner [5], Thomas Skurk [1,5] and Roman Lang [2,*]

[1] ZIEL—Institute for Food & Health, Technical University of Munich, Gregor-Mendel-Str. 2, 85354 Freising, Germany; beate.brandl@tum.de (B.B.); skurk@tum.de (T.S.)

[2] Leibniz Institute for Food Systems Biology at the Technical University of Munich, Lise-Meitner-Str. 34, 85354 Freising, Germany; c.czech.leibniz-lsb@tum.de (C.C.)

[3] TUM Graduate School, TUM School of Life Sciences Weihenstephan, Technical University of Munich, Alte Akademie 8, 85354 Freising, Germany

[4] Bavarian Center for Biomolecular Mass Spectrometry (BayBioMS), TUM School of Life Sciences, Technical University of Munich, Gregor-Mendel-Straße 4, 85354 Freising, Germany; susanne.wudy@tum.de

[5] Institute of Nutritional Medicine, TUM School of Medicine and Health, Technical University of Munich, Georg-Brauchle-Ring 62, 80992 Munich, Germany; hans.hauner@tum.de

[*] Correspondence: r.lang.leibniz-lsb@tum.de

**Abstract:** Health-related nutritional human studies rely on the validity of dietary data provided by study participants. Reliable biomarkers for food intake help objectify data collected by food frequency questionnaires. They facilitate the monitoring of compliance with the study requirements, e.g., abstinence from food, help clean biased data, and remove non-compliant individuals. Biomarker candidates are often revealed by sophisticated metabolomics analyses of body fluids, e.g., urine or plasma, collected from case and control study populations. However, validation for using a biomarker candidate in real-life scenarios is seldomly executed. Coffee is a food item of high interest because of the abundance of bioactive compounds and the regularity of life-time consumption by a large part of the population. Coffee has been found to positively impact cardiovascular risk, type 2 diabetes, and cognitive decline. Coffee and its health implications, therefore, are of high interest. A suitable dietary biomarker for coffee consumption is desirable for the clear classification of study participants as coffee drinkers or non-coffee drinkers to enable correlation of physiological response to dietary habits, e.g., coffee consumption. Here, we propose the roast coffee compound N-methylpyridinium (NMP) as a promising biomarker of pragmatic use to distinguish a coffee drinker from a non-coffee drinker. NMP is an easily accessible analytical target from the plasma and urine matrix that can help determine precedent exposure to roasted coffee products. We review the published information on the coffee compound N-methylpyridinium in foods, coffee, and plasma/urine after coffee consumption, and evaluate the data in the context of the proposed food biomarker criteria "plausibility", "time- and dose–response", "robustness", "reliability", "stability", "analytical performance", and "reproducibility". An additional data set is acquired to fill the gaps in the literature. In summary, we conclude that the abundance of NMP can serve as a reliable analytical tool to verify recent consumption of roasted coffee. The use of NMP appears limited to being qualitative, as NMP abundance in coffee and human biosamples is affected by several parameters, e.g., the roasting conditions and the volume and time of coffee consumed.

**Keywords:** N-methylpyridinium; roasted coffee; biomarker for food intake; dietary assessment

## 1. Introduction

Millions of people worldwide enjoy coffee as an integral aspect of their lifestyle. In the United States, 74% of people aged ≥20 years identify as coffee drinkers [1]. Survey data of the National Coffee Association (*n* = 2838, male and female) collected in 2020 indicate that

47–68% of adults $\geq$18 years consume coffee brew on a daily basis. The highest incentives for consuming 2–3 cups per day are hedonic, habitual, and stimulating reasons. This picture is similar in other countries. In 2022, European countries had a calculated per capita consumption of roasted coffee per year ranging from ~4 kg in Italy to ~10 kg in Luxemburg, making roasted coffee one of the most appreciated hot beverages [2–4].

Coffee is constantly in the focus of scientific research because of the high prevalence of its consumption and the impact of the brew on, e.g., mental health [5], physical and cognitive performance [6,7], emotions [8], and overall health [9]. In particular, the putative coffee-derived health benefits and hazards concern national healthcare systems, the coffee industry, and the individual consumer. Research interests range from the brew's hedonic and social value to the bioactivity and health effects of the whole food item as well as individual compound classes, e.g., melanoidins [10], phenolic compounds [11], diterpenes [12], and alkaloids [13]. Nutritional studies aim to unravel coffee's putatively beneficial contribution against cancers [14], cardiovascular health and diabetes [14–16], and cognitive function [7], as well as what can be expected from lifelong regular coffee consumption.

To investigate the health effects of defined food items, it is essential to have reliable tools to classify study participants as consumers or non-consumers, e.g., coffee drinkers and non-coffee drinkers. Such nutritional information is usually collected with food frequency questionnaires, a convenient but time-consuming and error-prone tool for data acquisition. It is known that self-reporting of dietary habits is often inaccurate due to a recall bias. Further, misreporting can be intentional [17], e.g., in foods considered "unhealthy". Objective parameters termed "Biomarkers for Food Intake" (BFI), which indicate consumption of a specific food item and can be acquired by instrumental analysis, are supposed to overcome this problem [18–20].

Discovery of new food-specific biomarker candidates is regularly reported in the literature [21–30]. Usually, the discovery process involves sample collection, e.g., urine or plasma, of low and high-consuming populations followed by untargeted metabolomics, approaches using sophisticated techniques like nuclear magnetic resonance (NMR) or ultra high-performance liquid chromatography coupled to high-resolution mass spectrometry (UPLC-HR-MS) [31,32]. The data are cleaned by several computational steps to separate the features that significantly contribute to separating the two study populations. In the case of UPLC-HR-MS-data, features are usually ions characterized by retention time and exact mass/sum formula. The downstream steps of biomarker research include compound identification, method development for detection and quantification, application, and validation as a BFI.

N-methylpyridinium (NMP, Figure 1) is among the biomarker candidates for coffee consumption [30]. We propose this analyte as an easily accessible analytical target in human biofluids to determine precedent exposure to roasted coffee. Here, we close a knowledge gap to thoroughly verify NMP as a reliable BFI in plasma and human biofluids by reviewing the available data for coffee intake concerning the proposed BFI criteria [20].

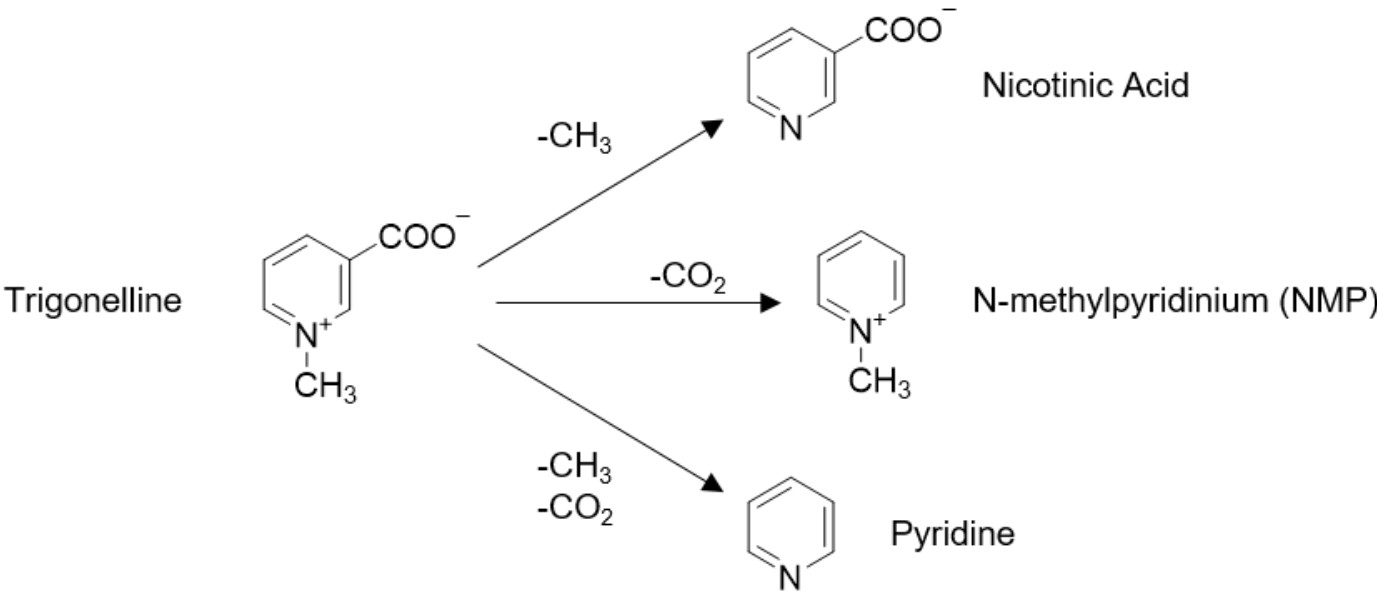

**Figure 1.** N-methylpyridinium (NMP) forms by heat-induced decarboxylation of the precursor trigonelline. Trigonelline is highly abundant in raw coffee and partly degrades during coffee roasting to generate NMP, minor amounts of nicotinic acid, and volatile pyridines.

## 2. Materials and Methods

All samples (plasma, dried blood spots, urine) were stored at −80 °C in the dark until analysis.

### 2.1. Criteria for the Validation of a Biomarker for Food Intake (BFI)

The proposed criteria for complete validation of a BFI are "plausibility", "dose–response", "time–response", "robustness", "reliability", "stability", "analytical performance", and "reproducibility" [20]. All available data on the formation of NMP in foods, its pharmacokinetic properties, and abundance in biofluids after exposure to coffee products will be discussed regarding these criteria. This approach was applied to obtain an overview of the current level of validation of NMP as a candidate BFI. Newly acquired quantitative data from cohort studies complemented the missing aspects.

### 2.2. Quantification of NMP

The publications of Lang et al. provide details of the analytical procedures [30,33–35]. Information on instrumental adaptations for acquiring additional data for this manuscript is in the Appendix A, cf. "Quantification of NMP". In brief, the biosample (25 µL) was mixed with a solution of the internal standard $d_3$-NMP (1 µM in acetonitrile/ethanol 9/1, 25 µL) and acetonitrile/ethanol (9/1, 250 µL). After vortexing and centrifugation (5 min, 4 °C, 13,520 rcf), the supernatant was transferred to an autosampler vial and analyzed by using UPLC-MS/MS on a hydrophilic interaction chromatography (HILIC) column with ammonium formate-buffered aqueous acetonitrile [30,33–35] (cf. Appendix A). The supernatant could be concentrated by evaporation and redissolution in acetonitrile/ethanol if concentrations were outside the calibrated range.

### 2.3. Samples of Plasma, Blood, Dried Blood Spots, and Urine

Quantitative NMP data were acquired in plasma and urine samples from the Enable study [36] for ROC-analysis, correlation of plasma and DBS data, and reproducibility among laboratories. Data from food frequency questionnaires were available. For the evaluation of analyte stability, plasma samples from a coffee intervention study from 2010 were used [33]. For ROC analysis of plasma, quantitative NMP data were linked to nutritional data ("coffee-drinker", *n* = 388, vs. "non-coffee drinker", *n* = 40, collected

in the FFQs). For ROC analysis of urine, quantitative urine NMP data of all the Enable study participants, who identified as "coffee drinker" in FFQ (*n* = 388) were analyzed versus all "non-coffee drinker" identifiers (*n* = 40). Food frequency data were collected as published [36,37]. These data are compiled in the Supplementary Data Excel file.

### 2.4. Statistical Analysis

Quantitative analysis of N-methylpyridinium (NMP) in biofluids (urine, plasma, dried blood spots) was performed by using LC-MS/MS and calculations were conducted in Analyst 1.6.3 (Sciex, Darmstadt, Germany). Data were further processed with Microsoft Excel 2016 and GraphPad Prism 9.3.0 for Windows (GraphPad Software, San Diego, CA, USA, www.graphpad.com). A receiver–operator characteristics (ROC) curve was calculated and plotted with GraphPad Prism 9.3.0 for Windows (Wilson/Brown Method). Zero values (NMP < lower limit of quantitation) were replaced by 0.01. Outlier removal from inter-laboratory analysis ("reproducibility") was performed by using ratio data Lab1/Lab2 using "ROUT" (Q = 1%, Graphpad Prism 9.3.0, cf. Appendix A Table A1).

### 3. Results

N-methylpyridnium (NMP) is a pyrolysis product of trigonelline [38]. It was isolated and identified as a compound unique to roasted coffee in 2002 [39,40]. NMP is detectable in the body fluids blood, urine, and saliva after coffee ingestion and, therefore, has been suggested as a Biomarker for Food Intake (BFI) indicating coffee consumption [30,34]. To validate the suitability of NMP as a BFI, the criteria "plausibility", "dose–response", "time–response", "robustness", "reliability", "stability", "analytical performance", and "re-producibility" as recently proposed [20] were evaluated. This analysis included published data on NMP in food and plasma after coffee exposure and additional data compiled in this manuscript (see below).

### 3.1. Additional Data from a Cross Sectional Study

For the comprehensive evaluation of NMP as a dietary biomarker for coffee intake, we needed to collect data from a free-living population, as quantitative NMP data in human biofluids are only available from interventions dedicated to coffee consumption, such as [30,33,34,41–43]. Therefore, we analyzed plasma, urine, and dried blood spots in sample sets from the Enable study, a cross-sectional study on nutritional habits without focus on coffee, cf. ref. [36]. Urine, plasma, and DBS samples were available from *n* = 461 adults (18–85 years) and urine samples from *n* = 44 children (3–5 years). Of the adults, *n* = 388 (84.2%) identified as coffee drinkers, *n* = 40 (8.7%) as non-coffee drinkers, and 33 individuals (7.1%) did not respond in this regard (Table 1).

**Table 1.** Descriptive data of NMP in plasma, dried blood spots, and urine in a free living population [a].

| | Coffee Drinkers (*n* = 388) | | | Non Coffee Drinkers (*n* = 40) | | | Children (*n* = 44) |
|---|---|---|---|---|---|---|---|
| | Plasma (nM) | DBS (nM) | Urine (µM) | Plasma (nM) | DBS (nM) | Urine (µM) | Urine (µM) |
| Mean (±SD) | 52.1 (±56.8) | 90.9 (±86.5) | 19.5 (±25.9) | 6.2 (±23.5) | 11.6 (±48.4) | 1.6 (±5.1) | 0.2 (±1.0) |
| Range (min–max) | 0.0–364.0 | 0.0–533.7 | 0.0–179.3 | 0.0–136.0 | 0.0–279.1 | 0.0–23.1 | 0.0–6.5 |
| Median | 37.2 | 73.7 | 12.9 | 0.01 | 0.01 | 0.01 | 0.0 |

[a] the study participants were grouped as "coffee drinker" and "non-coffee drinker", respectively, based on the food frequency questionnaire. Children (age 3–5 years) were considered non-coffee drinkers. Sample sets were available from the Enable study [36].

### 3.2. Plausibility

A BFI has to be specific for the respective food item. The N-methylpyridinium cation (NMP) is a prominent nonvolatile pyrolysis product of the coffee alkaloid trigonelline. It is formed by decarboxylation at temperatures >220 °C (Figure 1) [38,39]. NMP has been isolated from roasted coffee and found to be abundant in coffee powder and brew [40,44,45].

Its precursor, trigonelline, is a common compound in plant seeds [46]. However, the copious amounts found in coffee seeds are remarkable, ranging between 0.3 and 1.1% of the dry mass [47,48]. Trigonelline is present in Arabica and Robusta, the commercially relevant varieties. It is not removed by decaffeination or steam treatment. A substantial share degrades during the roasting process, giving rise to pyridines, nicotinic acid, and NMP [38,39]. There is a rough correlation between the coffee's roasting degree and its amount of NMP—the darker it is, the more NMP is contained [35,40]. A lightly roasted coffee contains ~500 μg/g, and a dark roasted espresso type coffee contains >2000 μg/g. NMP is water soluble and quantitatively extracted from the coffee powder into the brew when a reasonable (typical for coffee brew preparation) hot water/powder ratio ($\geq$16 $w/w$) is applied [35]. At least 20–40 mg/L of NMP is found in preparations from commercially available regular, mild, stomach-friendly, and decaffeinated coffee powders [45]. Trigonelline is common in plant material. However, coffee seeds are the only source in the human diet that is dry-heated to sufficiently high temperatures (190–260 °C) to generate NMP by decarboxylation. Compared to the copious amounts of NMP in roasted coffee seeds and the amounts of roasted coffee consumed daily, those present in, e.g., roasted fenugreek (*Trigonellum foenum greacum* L.) are neglectable [34,47,48]. NMP is a known metabolite of pyridine. Traces of pyridine have been found in various foods (cf. https://foodcomex.org/foodcomex_compounds/PC000192 (accessed on 20 September 2023) [49]). However, the concentrations of NMP formed by the metabolism of traces of pyridine can be expected to be negligible compared to the NMP concentrations in a single cup of coffee [50].

### 3.3. Time–Response

The pharmacokinetic properties of NMP were investigated in a dedicated study [33]. NMP was not detectable in plasma after a washout period of ten days. A single dose (350 mL) of roasted coffee brew was then consumed, and NMP was quickly taken into the bloodstream. NMP reached a plasma concentration Cmax of 757 $\pm$ 198 nM after 1–1.5 h with an estimated plasma elimination constant of ~0.3 (Figure 2A). The estimated plasma half-life ($t_{1/2}$) was ~2.15 h. Recent plasma data showed similar pharmacokinetic properties [41]. Urinary excretion started immediately after drinking coffee, and 62–69% of the ingested NMP was excreted within 8 h after coffee consumption [33]. In another coffee-drinking study [30], NMP/creatinine (nmol/μmol) ratios in morning urine samples from non-coffee drinkers, and coffee drinkers after omission of coffee consumption for four weeks, were ~0.1. A single dose of coffee brew led to an NMP/creatinine ratio of up to 10 in spot urine after 8 h and to ~0.1 after 240 h, when no further coffee was consumed (Figure 3A,B). Similar data were reported by Bakuradze et al. [42]. NMP in saliva increased from <0.1 μM to 100 μM upon coffee ingestion and quickly decreased, approaching ~0.1 μM already after 2 h [34].

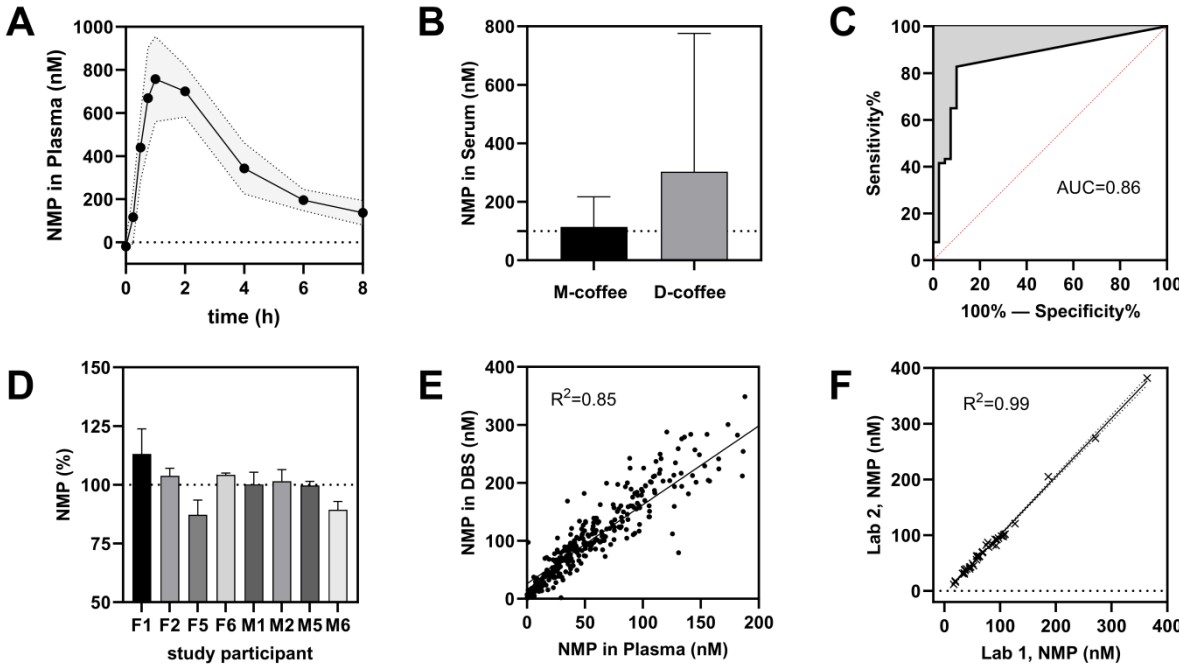

**Figure 2.** (**A**): NMP is bioavailable from a coffee brew with a plasma concentration peak ~1 h after coffee ingestion (data from [33], *n* = 13, f/m 7/6). Black line: mean, grey area: SD. (**B**): Consumption of roasted coffee containing 1.2 mg/g NMP ("D-coffee") and 0.39 mg/g NMP ("M-coffee"), respectively, leads to 303 ± 473 nM NMP and 114 ± 103 nM (mean ± SD) NMP, respectively, in serum (ratio coffee ~3, ratio serum 2.7, data from [43]). (**C**): Receiver–operator characteristics (ROC) curve of NMP in plasma (*n* = 388 coffee drinkers versus *n* = 40 non-coffee drinkers, data from food frequency questionnaires ([34], cf. Supplemental Excel file), cf. Appendix A Table A1). (**D**): NMP stability after storage at −80 °C, recovery rates from *n* = 8 samples: NMP data were reported in 2010 [33] and 2019 (cf. Appendix A Table A2). Panel (**D**) shows the recovery in % compared to recent measurements after storage at −80 °C in the dark for ~10 years. (**E**): The concentration of NMP in plasma and dried blood spots shows a good correlation ($R^2$ = 0.85, *n* = 329, *p* < 0.0001). DBS analysis, therefore, can give robust results in the more conveniently collected DBS sample compared to EDTA plasma prepared from venous blood. (**F**): Reproducibility across laboratories was tested. NMP was analyzed in two different laboratories with the same method. Linear regression from *n* = 39 pairs of values (5 outliers removed) gives $R^2$ = 0.99, *p* < 0.0001.

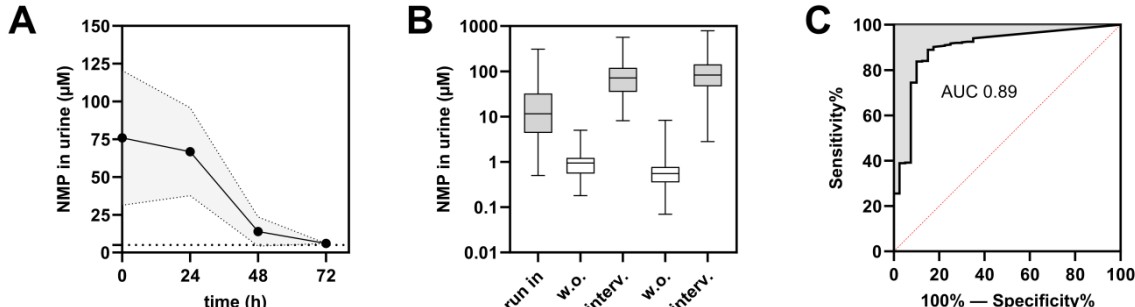

**Figure 3.** (**A**): After coffee consumption, NMP is excreted in the urine. NMP concentrations in morning spot urine are significantly higher in coffee drinkers' urine for 72 h compared to control urine (means ± SD, *n* = 9, black line: mean, grey area: SD, data from [30]). (**B**): NMP analysis has been used to document abstinence from coffee during washout periods (w.o.) and consumption of coffee during intervention periods (interv.) in health-related coffee intervention studies (means ± SD, *n* = 84, data from [40]). (**C**): NMP concentration in urine samples allowed discrimination of coffee non-consumers (*n* = 40) and coffee drinkers (*n* = 388), cf. Supplementary Excel file and Appendix A Table A5.

### 3.4. Dose Dependency

Kempf et al. (2015) investigated the cardiometabolic effects of differently roasted coffees on overweight adults [43]. The two coffee powders termed "M-coffee" and "D-coffee" contained $0.39 \pm 0.02$ mg/g and $1.20 \pm 0.03$ mg/g NMP (ratio "D-coffee"/"M-coffee" ~3, mean $\pm$ standard deviation). After a period of four weeks without coffee consumption, *n* = 114 participants were divided into two groups of 56 and 58 participants. Each group consumed 4–5 cups (125 mL/cup) of coffee daily for three months. NMP concentrations in serum were $114 \pm 103$ nM in the "M-coffee"-group and $303 \pm 473$ nM in the "D-coffee" group, suggesting dose dependency (ratio "D-coffee"-serum/"M-coffee"-serum ~2.7, mean $\pm$ standard deviation, Figure 2B). Bresciani et al. (2020) reported that the average plasma concentration (Cavg) and 24 h area under the curve (AUC 0–24 h) of NMP in plasma was roughly three times that when three cups of coffee were consumed over the day compared to consumption of a single cup [41]. Xi et al. (2021) recently reported that urinary NMP concentrations followed a dose–response dependency based on data from high- and low-coffee consumers [27].

### 3.5. Robustness

Plasma samples from a cross-sectional human study with complex meals were analyzed for NMP and compared with nutritional data from food frequency questionnaires [36] (Table 1). The mean plasma concentration in coffee drinkers (*n* = 388) was $52.1 \pm 56.8$ nM with a median concentration of 37.2 nM. Among the non-coffee drinkers (*n* = 40), we discovered three samples that were possibly misreported. Means ($6.2 \pm 23.5$ nM) and median (0.01 nM) concentrations were zero when these possible misreported samples were removed. The data from *n* = 388 coffee-drinking participants and *n* = 40 non-coffee-drinking adult individuals (including the three putative misreporters) were analyzed in a receiver–operator characteristic plot to characterize the differentiating power of NMP. As shown in Figure 2C, the plasma abundance of NMP allowed for robust discrimination of coffee drinkers and non-coffee drinkers (area under ROC curve = 86%). The area under ROC curve was 91% when the three putative misreporters were excluded (cf. Appendix A Table A1). Urine data showed that the area under ROC curve was 89% when coffee drinkers (*n* = 388) were compared to non-coffee drinkers (*n* = 40, Figure 3C, cf. Appendix A Table A5). When the urine samples from the same putatively misreporting individuals were removed, the area under the ROC curve was 94% (92.1–96.8, $p < 0.0001$).

An intervention study investigating the impact of coffee on energy intake and satiety reported NMP concentrations in spot urine. During the coffee periods, NMP concentrations were 10–100 µM (NMP/creatinine, nmol/µmol ~10), whereas during the washout periods, they ~1 µM (NMP/creatinine nmol/µmol ~0.1) (Figure 3B). Therefore, the NMP/creatinine was a robust indicator of compliance with the study protocol [42]. The additional data reported in this manuscript corroborate these findings with similar data (cf. Table 1).

### 3.6. Reliability

Several investigations reported NMP was absent from plasma and minimal in urine when no roasted coffee was consumed for several days [30,33,41,42]. Volunteers were restricted from coffee, cocoa, tea, and vitamin supplements for ten days in a study dedicated to evaluating the bioappearance of NMP and other pyridine derivatives [33]. NMP was undetectable in the samples taken before coffee consumption. After drinking a single dose (350 mL) of roasted coffee brew, an increase in NMP was detected. We assessed NMP in different matrices (DBS, plasma, urine) of samples from a cross-sectional human study (cf. Table 1) and correlated the results with the data ("coffee drinker", "non-coffee drinker") collected via a validated food frequency questionnaire (FFQ) [36]. Based on the plasma data, we could differentiate between coffee and non-coffee drinkers (Figure 2C, cf. Robustness, Appendix A Table A1).

*3.7. Stability*

　　We analyzed plasma samples from a coffee intervention study conducted in 2010 [33] after storage at −80 °C to investigate the stability of NMP in the matrix. The precision of replicates (*n* = 3) was generally good, with <9.5% RSD. Comparing the concentrations of NMP of the same samples analyzed in 2010 and 2019, we found deviations ≤13% (Figure 2D). Despite the small sample size, we conclude that no substantial concentration changes of NMP in plasma occurred within these ~ten years and that re-analysis of samples delivers robust results (Appendix A Table A2).

*3.8. Analytical Performance*

　　NMP showed no substantial retention on standard reversed phases, but successful chromatographic separation has been achieved on cation exchange columns with buffered eluents [39,40,45]. The use of HILIC columns has been further developed for analyzing the compounds in coffee and the biomaterials urine, saliva, serum, and plasma [30,33–35,42,43]. The cationic nature of N-methylpyridinium ($m/z$ 94) suggests detection by mass spectrometry. The compound produces the stable pyridinium ($m/z$ 79) fragment, which forms by collision-induced removal of the nitrogen-bound methyl group [40]. A stable isotopologue as the internal standard for stable isotope dilution analysis (SIDA) can easily be synthesized from pyridine and deuterated iodomethane and subsequent crystallization of the iodide salt from acetonitrile [40]. The quantitative analysis by SIDA showed consistent performance due to the use of $d_3$-NMP as the internal standard, effectively compensating for losses during sample preparation [30,33–35,42–45]. Method development experiments with matrix-free calibration standards (solvent) and standards diluted in matrix showed good linear response in the mass spectrometer. Calibration in a solvent was sufficient for accurate quantitation in coffee brew, urine, saliva, and plasma, as recovery experiments indicate. The following table summarizes the analytical performance of stable isotope dilution analysis (SIDA) of NMP in different matrices (Table 2, Appendix A Tables A3 and A4, and Appendix A Figure A1).

**Table 2.** Analytical performance of stable isotope dilution assays for quantification of N-methylpyridinium in coffee and biological fluids (urine, saliva, plasma, blood).

| Matrix | $R^{2\ a}$ | Precision (RSD, %) | Accuracy (%) | Precision (RSD, %) | Accuracy (%) | Prec. of Authentic Samples (%) | Ref. |
|---|---|---|---|---|---|---|---|
| | | **Calibration Standards** | | **Quality Controls (Spiked Matrix)** [b] | | | |
| Solvent | >0.999 | <11 | 97–102 | <11.9 (hum. Plasma) | 97–103 (hum. Plasma) | 3–6.5 (coffee brew) | [33,35] |
| Urine [c] | >0.999 | <3 | 91–105 | <3 | 98–102 | | [30] |
| Saliva | >0.99 | <14 | 94–103 | <3 | 99–100 | | [34] |
| Plasma [d] | 0.999 | ≤5.1 | 85–109 | ≤2.7 | 96–107 | | SI |
| Blood [e] | 0.999 | ≤8.4 | 91–111 | ≤1.9 | 105–113 | | SI |

[a] linear regression, [b] analyte-free matrix spiked with the analyte, [c] artificial urine, [d] human EDTA plasma, [e] porcine EDTA blood. Confer Appendix A Figure A1.

　　Moreover, dried bloods pots (DBS) are convenient for storing small sample amounts and handling samples, e.g., postal transportation for analysis. In the samples from the same cross-sectional study [36], we analyzed human plasma and DBS collected (DBS and plasma sample sets available from *n* = 329) to compare the values and evaluate the suitability of DBS for NMP analysis. The data showed a robust correlation ($R^2$ = 0.85, *p* < 0.0001), suggesting that the DBS analysis of NMP delivers trustworthy information (Figure 2E).

*3.9. Reproducibility*

To investigate inter-laboratory reproducibility, *n* = 44 plasma samples from the Enable study [36] were analyzed for NMP in two different laboratories. The data acquired at the Leibniz Institute for Food Systems Biology (LSB) at the Technical University of Munich and at the Bavarian Center for Biomolecular Mass Spectrometry (BayBioMS) at the Technical University of Munich showed excellent correlation ($R^2$ = 0.99, $p < 0.0001$). After the removal of outliers (*n* = 5), the ratios of the acquired concentrations fell between 90 and 127%. The results are correlated in Figure 2F (cf. Supplementary Data in Excel file).

## 4. Discussion

Dietary biomarkers play a crucial role in nutrition and food research, enabling scientists to identify and measure indicators that provide valuable information on food quality, safety, and health.

Reliable biomarkers that indicate coffee consumption are highly interesting for health-related nutritional studies, and several metabolomics studies have been dedicated to discovering suitable compounds [26–30,51]. However, whether such proposed candidates are actually applied for biomarker purposes is still being determined. Promising candidates, e.g., phase II metabolites of hydroxycinnamic acids [11,28], furane metabolites [51], and diterpenes [26,29], are not readily available as standard substances or have yet to be sufficiently validated. Using a profile of compounds has been suggested to increase accuracy for classifying study participants [11,27]. However, in real-life scenarios, analyzing and processing multiple compounds per individual to verify or substitute dietary records appears cumbersome, making a quick classification of an individual as a coffee drinker or a non-coffee drinker impossible.

Among the biomarker candidates for coffee exposure, we suggest NMP as a reliable and robust measure for application in real-world scenarios. NMP is a coffee-specific compound that shows appreciable properties as a biomarker for exposure to roasted coffee. Its substantial abundance in coffee is directly linked to roasting raw coffee seeds at exceptionally high temperatures (>220 °C), a unique processing method for a food item. NMP is present in both roasted Arabica and Robusta regardless of treatments like steaming for stomach-friendly coffees or decaffeination. Its bioavailability, pharmacokinetic profile in plasma, and excretion profile are well established. NMP is excreted without any known substantial metabolism-related changes.

NMP is detectable in blood, serum [43], plasma [33,41], saliva [34], and urine [27,30,41,42]. However, the determined NMP concentration in, e.g., urine does not allow for an estimation of how much coffee brew has been ingested because of several reasons. The excretion characteristics of a compound can be affected by multiple parameters, e.g., physical activity, time-point of sample collection, and consumption of water and other beverages throughout the day. Further, as detailed above, the abundance of NMP in the respective coffee brew is strongly affected by the roasting degree of the coffee, the abundance of its precursor trigonelline in the raw coffee, the water/coffee powder ratio, and the consumed volume of coffee [35]. Therefore, there is yet no concept to translate an NMP concentration measured in a biosample into a consumed coffee volume or a specific time-point when the coffee was consumed. However, the accumulated data presented here suggest that NMP can serve as a robust qualitative marker compound, e.g., for monitoring compliance in coffee intervention trials [42].

In contrast to complex conjugated metabolites or even profiles of several metabolites, analysis of NMP as a single marker bears substantial advantages. In particular, the fact that results are quickly obtained is valuable in clinical large-scale study settings for real-time monitoring or simple compliance control. It is less expensive than measuring multiple analytes with regard to the acquisition of reference compounds, equipment maintenance, reagents and disposables, analysis time, and human resources for machine and data handling.

### 4.1. Plausibility

N-methylpyridinium is a pyrolysis product of trigonelline generated during coffee roasting [35,38–40]. Coffee roasting requires exceptionally high temperatures compared to other heating methods, e.g., boiling, used in food processing. When coffee consumption is omitted, NMP is no longer detectable in plasma [33], and only traces are found in urine [30,42] (Figure 3B). Therefore, NMP, as a roasting product of trigonelline, seems to be highly specific for roasted coffee intake, and the criterion "plausibility" is fulfilled.

### 4.2. Time–Response and Dose–Response

Consumption of coffee brew leads to a rapid increase of NMP in plasma in a time-dependent manner. Pharmacokinetic data in plasma/serum and urine (Cmax, tmax, λ1/2) after single and repeated exposure have been published [27,30,33,41,42] and indicate dose dependency [27,41,43]. The criteria "time–response" and "dose–response" are met in both plasma and urine.

### 4.3. Robustness

Quantitative data for the abundance of NMP in urine [27,30,33,34,41,42], plasma [33,41], saliva [34], and serum [43] after coffee exposure have been reported in several trials. We here add new data from a cross-sectional study with complex meals as suggested for the biomarker validation criterion "robustness". This study proved the uniqueness of NMP as a BFI under free-living conditions [36]. The abundance of NMP in plasma and urine discriminated between coffee and non-coffee drinkers when consuming an individual diet (area under the ROC curve = 0.86 and 0.89, Figures 2C and 3C). Therefore, the criterion "robustness" is fulfilled.

### 4.4. Reliability

A biomarker needs to be reliable for a specific condition. When coffee consumption was omitted, NMP was not detectable in plasma [33,41]. After coffee consumption, plasma levels steeply increased, indicating NMP was taken up from the food. In the present investigation, the FFQ data reflected the NMP results acquired in samples from the Enable study [36]. The AUC of the ROC curve was 0.86 in plasma and 0.89 in urine (Figures 2C and 3C). NMP concentrations in urine from non-coffee drinkers were minimal (~1 µM, Table 1) and reached 10–600 µM when coffee was consumed ([30,42], Figure 3B). We conclude that the criterion "reliability" of NMP in plasma and urine for "specific to coffee consumption" is fulfilled.

### 4.5. Stability

The stability of the biomarker in a given sample type needs to be documented. The data presented here show that the concentrations of NMP determined in coffee drinkers' plasma samples after ~ten years of storage at $-80\,^\circ$C gave the same results ($\leq 13\%$ deviation) as before ([33], Figure 2D, Appendix A Table A2). Therefore, the criterion "stability" is fulfilled for the matrix plasma.

### 4.6. Analytical Performance

The quantification of NMP was successfully conducted by stable isotope dilution analysis, the golden standard in quantitative analysis, in all matrices relevant to biomarker analysis (blood, plasma, dried blood spots, saliva, and urine, cf. Table 1 and references therein). Therefore, the criterion of "analytical performance" is fulfilled.

### 4.7. Reproducibility

Reproducibility describes when two laboratories analyze the same sample and obtain comparable results. Analyses of NMP in a subset of $n = 44$ plasma samples in two different laboratories and LC-MS/MS systems under the same conditions gave similar results, as

indicated by a correlation coefficient of $R^2 = 0.99$, $p < 0.0001$ (Figure 2F). The criterion of "reproducibility" for plasma samples stored at $-80$ °C is fulfilled.

## 5. Conclusions

We compiled and reviewed the literature data to validate NMP as a dietary biomarker concerning the proposed criteria of "plausibility", "dose–response", "time-response", "robustness", "reliability", "stability", "analytical performance", and "reproducibility." These criteria were examined based on a literature analysis and newly acquired data. We conclude that abundance of NMP can serve as a reliable analytical tool to verify recent consumption of roasted coffee. The use of NMP appears limited to being qualitative, as NMP abundance in coffee and human biosamples is affected by several parameters, e.g., the roasting conditions and the volume and time of coffee consumed.

**Supplementary Materials:** The following supporting information can be downloaded at: https://www.mdpi.com/article/10.3390/beverages10010012/s1, Excel data file with raw data (Supplementary Data Excel file).

**Author Contributions:** Conceptualization, R.L., B.B. and T.S.; methodology, R.L., C.C. and S.I.W.; formal analysis, R.L. and B.B.; investigation, C.C., S.I.W. and A.B.; data curation, R.L. and B.B; writing—original draft preparation, R.L., B.B. and T.S.; writing—review and editing, R.L., B.B., H.H. and T.S. All authors have read and agreed to the published version of the manuscript.

**Funding:** This research received no external funding.

**Data Availability Statement:** All data acquired and analyzed in this manuscript are supplied in the Appendix A and the supporting information data file.

**Acknowledgments:** We acknowledge Dorothee Volkert, Institute for Biomedicine of Aging, Friedrich-Alexander-Universität Erlangen-Nürnberg, Nuremberg, Germany for valuable discussions. The Graphical Abstract was created with Biorender (www.biorender.com, accessed on 10 January 2024).

**Conflicts of Interest:** The authors declare no conflicts of interest.

## Appendix A

Chemicals and Materials—N-methylpyridinium iodide (NMP) and $d_3$-N-methylpyridinium iodide ($d_3$-NMP) were synthesized from pyridine and iodomethane, and pyridine and $d_3$-iodomethane, respectively, and crystallized from acetonitrile. NMR and MS data were consistent with previous reports [39]. Blank human urine and plasma were obtained from Hölzel Diagnostika Handels GmbH (Köln, Germany). Porcine blood (EDTA) was a generous gift from Dr. Habil. Krzysztof Flisikowski (Chair of Livestock Biotechnology, TUM School of Life Sciences, Freising, Germany). The eluent for UPLC-MS/MS analysis was prepared as follows: In a measured flask (1 L), aqueous ammonium acetate (5 M in water, 1 mL) was mixed with Millipore water (109 mL) and formic acid (98%, 15 mL). The mixture was filled up to the final volume with acetonitrile (LC-MS quality).

Quantification of NMP—Stock solutions and calibrations. Stock solutions of $d_3$-NMP iodide (internal standard, IS) and NMP iodide (analyte) were prepared in acetonitrile/ethanol 9/1 at a concentration of 1 µM. The analyte stock was diluted in 1 + 1 steps with water/methanol (80/20, $v/v$) to yield the concentrations 50, 25, 12.5, 6.25, 3.13, 1.56, 0.78, 0.39, and 0.19 µM. Aliquots of the dilutions were spiked into human blank plasma, porcine blood, and water (100 µL dilution + 900 µL matrix) to yield matrix calibration samples with an NMP concentration ranging from 5 to 0.019 µM. Additional similarly prepared matrix samples served as quality controls for validation. Dried blood spots were prepared by pipetting an aliquot of spiked blood (25 µL) onto protein saver cards (903 Protein Saver Card, Whatman, GE Healthcare) and subsequent air-drying overnight.

Quality controls (QC)—QC samples were prepared in duplicates in blank human EDTA-plasma and porcine EDTA-blood at low (156 nM) and high (1250 nM) concentrations. QCs were analyzed in triplicates.

Sample preparation— Matrix calibration standards and authentic samples (urine, blood, plasma) were prepared as follows: The sample (25 μL) was spiked with the IS-solution (1 μM, 25 μL), vortexed, and diluted with acetonitrile/ethanol (9/1, *v/v*, 250 μL). The suspension was vortexed, centrifuged (5 min, 4 °C, 13,520 rcf), and the supernatant was transferred to an HPLC vial with 200 μL –insert. Aliquots (1 μL) were injected into the UPLC-MS/MS system. DBS were cut out, placed in Eppendorf caps, spiked with the internal standard solution (25 μL), and extracted with acetonitrile/ethanol (9/1, *v/v*, 1 mL) with sonication. After centrifugation (5 min, 4 °C, 13,520 rcf), an aliquot of the supernatant (1 μL) was injected into the UPLC-MS system. Separation was achieved using the same conditions as reported earlier.

Instrumentation—The UPLC (Exion UPLC, Sciex) was connected to a 6500+ triple quad mass spectrometer (Sciex, Darmstadt, Germany). Chromatography was achieved on a HILIC Kinetex column (50 × 2 mm, 2.1 μm, Phenomenex, Aschaffenburg, Germany) with isocratic elution at a flow rate of 0.5 mL/min [30]. The MS/MS system operated in positive electrospray. Ion source temperature was 500 °C, ion spray voltage (IS): 5.5 kV, curtain gas: 35 psi, collision gas: medium, nebulizer Gas: 50 psi, drying Gas: 60 psi. Ion path parameters of the analyte NMP iodide and the internal standard $d_3$-NMP iodide were tuned using software-assisted optimization. NMP: Q1 93.9 ($[M]^+$); Q3 78.9, 51.9; declustering potential (DP) 46, 46; entrance potential (EP) 10, 10; collision energy (CE) 35, 43; cell exit potential (CXP) 32, 24. $d_3$-NMP: Q1 97.0 ($[M]^+$); Q3 78.9, 51.9; declustering potential (DP) 56, 56; entrance potential (EP) 10, 10; collision energy (CE) 27, 45; cell exit potential (CXP) 10, 24.

**Table A1.** Data (means ± standard deviation and median concentrations) on NMP in plasma from coffee drinkers and non-coffee drinkers (Figure 2C). Area under ROC curve, standard error, 95% confidence interval and *p* value.

| | Coffee Drinkers | Non Coffee Drinkers | Non-Coffee Drinkers without Misreporters |
|---|---|---|---|
| *n* | 388 | 40 | 37 |
| c (nM) | 52.0 ± 57.4 | 6.2 ± 23.4 | 0 |
| c min (nM) | 0.0 | 0.0 | 0 |
| c max (nM) | 364.0 | 135.9 (three outliers, possibly misreported: 43.6 nM, 135.9 nM, 47.1 nM) | 0 |
| Median | 37.2 | 0.01 | 0.01 |
| Area under ROC curve | | 0.8617 | 0.9065 |
| Std. Error | | 0.0299 | 0.0157 |
| 95% confidence interval | | 0.8030–0.9203 | 0.8757–0.9373 |
| *p* value | | <0.0001 | <0.0001 |

**Table A2.** Stability of NMP in plasma after storage at −80 °C for ~ten years (2010–2019, [33]). (Figure 2D).

| Sample | C NMP (Means ± SD) | | Ratio |
|---|---|---|---|
| | 2010 [a] | 2019 (*n* = 3) [b] | |
| 1 | 839.0 ± 134.4 | 949.0 ± 89.2 | 1.13 |
| 2 | 673.5 ± 33.3 | 698.9 ± 21.8 | 1.04 |
| 3 | 872.5 ± 37.5 | 760.5 ± 55.2 | 0.87 |
| 4 | 741.5 ± 31.8 | 772.8 ± 5.6 | 1.04 |
| 5 | 1287.0 ± 4.2 | 1288.8 ± 66.7 | 1.00 |
| 6 | 760.0 ± 101.9 | 771.3 ± 37.6 | 1.01 |
| 7 | 537.0 ± 55.2 | 535.0 ± 9.8 | 1.00 |
| 8 | 586.0 ± 53.7 | 523.3 ± 20.9 | 0.89 |

[a] Lang et al., 2010 [33], [b] this paper.

**Table A3.** Calibrated range, precision, accuracy, and $R^2$ of stable isotope dilution analysis of biomarker candidate N-methylpyridinium in solvent, plasma, and blood.

| Matrix | Calibration Standards [a] | | | |
|---|---|---|---|---|
| | Calibrated Range (nM) | Precision (RSD, %) | Accuracy (%) | $R^2$ |
| Solvent | 19–10,000 | ≤4.9 | 95–102 | 0.995 |
| Plasma (EDTA) | 19–10,000 | ≤5.1 | 85–109 | 0.999 |
| Blood (porcine, EDTA) | 19–5000 | ≤8.4 | 91–111 | 0.999 |
| Dried Bloodspot (porcine, EDTA) | 19–5000 | ≤10.2 | 64–97 | 0.998 |

[a] means of triplicate injections.

**Table A4.** Precision and accuracy of stable isotope dilution analysis of quality controls of biomarker candidate N-methylpyridinium in solvent, plasma, and blood.

| Matrix | Low QC (156 nM) | | | High QC (1250 nM) | | |
|---|---|---|---|---|---|---|
| | Found (nM) | Precision (RSD, %) | Accuracy (%) | Found (nM) | Precision (RSD, %) | Accuracy (%) |
| Solvent (20% aq. MeOH) [a] | 154.0 ± 7.6 | 5.0 | 98.7 | 1215.0 ± 23.6 | 1.9 | 97.2 |
| Plasma (EDTA) [a,b] | 167.9 ± 4.6 | 2.7 | 107.6 | 1204.7 ± 15.5 | 1.3 | 96.4 |
| | 157.9 ± 4.6 | 2.9 | 101.3 | 1378.7 ± 12.5 | 0.9 | 110.3 |
| Blood (porcine, EDTA) [a,b] | 173.2 ± 3.4 | 1.9 | 111.0 | 1308.0 ± 14.9 | 1.1 | 104.6 |
| | 175.8 ± 2.6 | 1.5 | 112.7 | 1350.4 ± 25.1 | 1.9 | 108.0 |

[a] data are means ± standard deviation of $n$ = 3; [b] QC samples were prepared in duplicates and analyzed in triplicates.

**Table A5.** NMP in human urine. Area under ROC curve, standard error, 95% confidence interval, and $p$ value.

| | Coffee Drinkers | Non-Coffee Drinkers |
|---|---|---|
| $n$ | 388 | 40 |
| c (µM) | 19.5 ± 25.9 | 1.6 ± 5.1 |
| c min (µM) | 0.0 | 0.0 |
| c max (µM) | 179.3 | 23.1 |
| Median | 12.9 | 0.01 |
| Area under ROC curve | 0.8996 | |
| Std. Error | 0.0276 | |
| 95% confidence interval | 0.8455–0.9538 | |
| $p$ value | <0.0001 | |

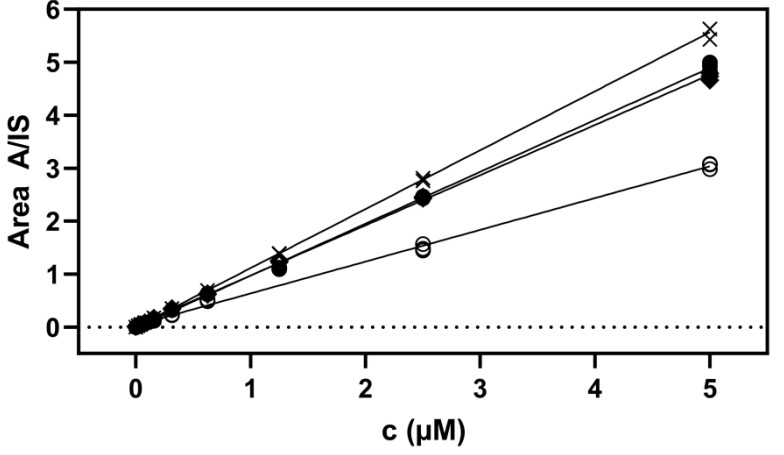

**Figure A1.** Matrix calibration of NMP in solvent, plasma, blood, and as dried blood spots (Concentration-Area (analyte/internal standard)-plots of NMP) show good linearity.

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
