# Peer review of "Validation of N-Methylpyridinium as a Feasible Biomarker for Roasted Coffee Intake"

_beverages, doi:10.3390/beverages10010012_

Round 1
Reviewer 1 Report
Comments and Suggestions for Authors
Thank you for allowing me to read this interesting and well-prepared manuscript. I feel that this is useful for other researchers in terms of providing a rationale for their choice of assays. I only have a couple of points that the authors may wish to consider…
Abstract:
Lines 31-32: The authors could consider including the ergogenic effect in a sporting / exercise context of coffee ingestion.
There does not appear to be a solid conclusion in the abstract. Some along the lines of ‘We conclude that abundance of NMP can serve as a reliable analytical tool to verify recent consumption of roasted coffee. The use of NMP appears limited to being qualitative, as NMP abundance in coffee and human biosamples is affected by several parameters, e.g., the roasting conditions and the volume and time of coffee consumed.’ as presented in the conclusion would be more suitable.
Introduction:
Overall, this provides a strong rationale for the study. However, it might be useful to strengthen the reasoning for why researchers might want to be aware if the participants have recent consumption of coffee.
Author Response
Comment Thank you for allowing me to read this interesting and well-prepared manuscript. I feel that this is useful for other researchers in terms of providing a rationale for their choice of assays. I only have a couple of points that the authors may wish to consider…
Answer Dear reviewer, many thanks for your kind words and your efforts to improve the manuscript.
Comment Lines 31-32: The authors could consider including the ergogenic effect in a sporting / exercise context of coffee ingestion.
Answer Thank you for the suggestion. We include performance enhancement with reference [4].
Comment There does not appear to be a solid conclusion in the abstract. Some along the lines of ‘We conclude that abundance of NMP can serve as a reliable analytical tool to verify recent consumption of roasted coffee. The use of NMP appears limited to being qualitative, as NMP abundance in coffee and human biosamples is affected by several parameters, e.g., the roasting conditions and the volume and time of coffee consumed.’ as presented in the conclusion would be more suitable.
Answer Many thanks for your suggestion. We did as suggested (lines 42-46): “In summary we conclude that the abundance of NMP can serve as a reliable analytical tool to verify recent consumption of roasted coffee. The use of NMP appears limited to being qualitative, as NMP abundance in coffee and human biosamples is affected by several parameters, e.g., the roasting conditions and the volume and time of coffee consumed.”
Comment Introduction: Overall, this provides a strong rationale for the study. However, it might be useful to strengthen the reasoning for why researchers might want to be aware if the participants have recent consumption of coffee.
Answer Thank you for the suggestion. We added “A suitable dietary biomarker for coffee consumption is desirable for clear classification of the study participants as coffee drinkers or non-coffee drinkers to enable correlation of physiological response to dietary habits, e.g., coffee consumption” in lines 33-35.
Reviewer 2 Report
Comments and Suggestions for Authors
This is an interesting and nice study perfectly matching the scope of the journal and having apparent practical importance. It is well written and makes good impression. I recommend it for publication after resolving some important issues detected when reading the manuscript:
1. Do the authors have any explanation of the high NMP contents in the samples taken from Non coffee drinkers (Table 1)? Why the urine of non-coffee drinkers contained higher maximum amounts of NMP (279.1 uM) when compared to coffee drinkers (179.3 uM)? Also, I could not understand why the median value in this case was 0.
2. It is known that trigonelline is widespread in foods. Do the authors have any information about the possibility of the formation of NMP from other dietary sources e.g. bread crust or fried potatoes, which are also exposed to significant temperatures? Can other sources be responsible for NMP in non-coffee drinkers’ samples?
3. From the practical point of view, the thorough description of analytical methods for NMP quantification is crucial. Thus, I suggest introducing an additional Table summarizing the key analytical parameters (chromatographic conditions, detection technique, linear range, and LODs/LOQs) of the existing methods. The attained limits of detection and quantification should be further discussed in terms of their sufficiency for reliable detecting NMP in blood, urine and saliva.
4. Some discussion on the limitations of the NMP analysis for the discrimination of coffee and non-coffee drinkers should be provided.
5. Conclusion section seems not informative. Please, add some key data as concentrations, period after coffee consumption in which NMP may be detected, existing limitations, etc.
Author Response
Comment This is an interesting and nice study perfectly matching the scope of the journal and having apparent practical importance. It is well written and makes good impression. I recommend it for publication after resolving some important issues detected when reading the manuscript:
Answer Dear Reviewer, many thanks for your kind words and your efforts to improve the manuscript. We have considered all comments and repond accordingly (see below).
Comment Do the authors have any explanation of the high NMP contents in the samples taken from Non coffee drinkers (Table 1)?
Answer Dear reviewer, many thanks for pointing this out. There apparently is a typo in table 1. As can be found in the Excel-file that provides all data, sheet “data sorted coffee – non coffee”, columns D (coffee) and J (non-coffee) lines 8 (maximal observed value) the urine concentration cmax in non-coffee drinkers is 23.1 µM (as opposed to 179.3 µM in coffee drinkers). The typo is corrected and in line with the provided raw data in the Excel file.
Comment Why the urine of non-coffee drinkers contained higher maximum amounts of NMP (279.1 uM) when compared to coffee drinkers (179.3 uM)? Also, I could not understand why the median value in this case was 0.
Answer Please see the answer above. The non-coffee data for urine and dried-blood-spots (DBS) were falsely interchanged. Non-coffee drinkers’ urine contained a maximum of 23.1 µM, that of coffee drinkers 179.3 µM. Please note that urine concentrations are given in µM while the concentration in plasma and DBS is given in nanomolar (nM). We corrected the median.
Comment It is known that trigonelline is widespread in foods. Do the authors have any information about the possibility of the formation of NMP from other dietary sources e.g. bread crust or fried potatoes, which are also exposed to significant temperatures? Can other sources be responsible for NMP in non-coffee drinkers’ samples?
Answer The reference [30] investigates other potential dietary sources for NMP, but the list does not include fried potato. The topic is discussed in lines 179-187. As given in the foot-note of table 1, the study participants were classified as “coffee-drinkers” and “non-coffee drinkers” based on their answers in the "gold standard food frequency questionnaire (FFQ)", not based on the presence of NMP. We suggest those samples containing NMP despite being “non-coffee” are misreported (line 223-231), underlining the necessity for application of dietary biomarker analysis.
Comment From the practical point of view, the thorough description of analytical methods for NMP quantification is crucial. Thus, I suggest introducing an additional Table summarizing the key analytical parameters (chromatographic conditions, detection technique, linear range, and LODs/LOQs) of the existing methods. The attained limits of detection and quantification should be further discussed in terms of their sufficiency for reliable detecting NMP in blood, urine and saliva.
Answer The method, LC conditions and parameters are provided in references [28,29,30,38,39,41] and are detailed in Appendix A1 (lines 439-484).
Comment Some discussion on the limitations of the NMP analysis for the discrimination of coffee and non-coffee drinkers should be provided.
Answer Limitations are discussed in lines 349-361 and stated in 422-426. The main limitations are the quick excretion (line 351), the variability of NMP in the brew (e.g., roasting degree, line 354) and the variable individually consumed amount (3-5 cups per day, line 357). We therefore conclude NMP could only serve as a qualitative biomarker (line 360, line 424).
Comment Conclusion section seems not informative. Please, add some key data as concentrations, period after coffee consumption in which NMP may be detected, existing limitations, etc.
Answer The key data on concentration, period after coffee consumption in which NMP can be detected, existing limits etc. are reported in the Results and discussed in the Discussion according to the authors guidelines of Beverages.
While the Results section is structured according to the suggested validation criteria (cf. [18]) to compile NMP literature data, the Discussion section discusses the individual criteria in the context of applicability of NMP as a dietary biomarker for coffee consumption. Limitations of NMP as a dietary biomarker are stated in lines 349-361. The Conclusion summarizes the manuscript as it answers the question: “what can I expect from using NMP as a biomarker for coffee consumption”.
Reviewer 3 Report
Comments and Suggestions for Authors
This is a review paper on the measurement of N-methylpyridinium in
biological samples as a biomarker of coffee intake.
The overall content and text structure are well organized and easy to read.
A description of the storage conditions of the specimen and the stability
of the substance is required. L106: 1200 rpm is a meaningless number depending on radius.
The value converted to gravity should be recorded. Variations in the notation of AUC and AUROC should be unified. Correlation coefficients must be shown together with P values. The explanation of Fig. 2 is insufficient. What does the gray area in A) indicate? BD) is the mean and SD? C) AUC requires a 95% confidence interval and P value. EF) also requires a P value for the R2 value.
The conclusion is enough designed considering the reviewed studies. The references were appropriate.
Author Response
Comment This is a review paper on the measurement of N-methylpyridinium in biological samples as a biomarker of coffee intake. The overall content and text structure are well organized and easy to read.
Answer Dear reviewer, many thanks for your kind words and your efforts in improving the manuscript.
Comment description of the storage conditions of the specimen and the stability of the substance is required.
Answer Thank you for the suggestion. We added the information on storage of specimen (line 99-100, line 316). Information on the stability of N-methylpyridinium in biological samples is not available in the literature. However, we answer this question for plasma in the present manuscript (cf. section 3.7 and line 316).
Comment L106: 1200 rpm is a meaningless number depending on radius. The value converted to gravity should be recorded.
Answer Thank you for the suggestion. We changed the value accordingly (lines 114, 471, 475).
Comment Variations in the notation of AUC and AUROC should be unified.
Answer Thank you for the suggestion. For clarification we referred to “Area under the ROC curve” (see lines 232, 234, Appendix Tables A1 and A5).
Comment Correlation coefficients must be shown together with P values.
Answer We added P values (Lines 291, 303, 317, 321, Appendix Tables A1 and A5)
Comment The explanation of Fig. 2 is insufficient. What does the gray area in A) indicate?
Answer Thank you. The grey area marks the standard deviation. We have revised the figure legend.
Comment BD) is the mean and SD?
Answer Thank you. Yes, it is the mean±standard deviation. We have updated the figure legend.
Comment C) AUC requires a 95% confidence interval and P value. EF) also requires a P value for the R2 value.
Answer Thank you. We report the respective data for the ROC curves in Appendix Table A1 and Appendix Table A5, and reference the tables in lines 233 and 329, respectively.
Comment The conclusion is enough designed considering the reviewed studies. The references were appropriate.
Answer Thank you for your review.
Round 2
Reviewer 2 Report
Comments and Suggestions for Authors
The authors addressed my most serious concerns regarding the data presented in Table 1. Now I can recommend the manuscript for publication.